# Effects of Changing Temperature on Gross N Transformation Rates in Acidic Subtropical Forest Soils

**Xiaoqian Dan [1], Zhaoxiong Chen [1], Shenyan Dai [1,\*], Xiaoxiang He [1], Zucong Cai [1,2], Jinbo Zhang [1,3] and Christoph Müller [4,5]**

[1] School of Geography, Nanjing Normal University, Nanjing 210023, China; 181302119@stu.njnu.edu.cn (X.D.); 181302002@stu.njnu.edu.cn (Z.C.); 191301031@stu.njnu.edu.cn (X.H.); zccai@njnu.edu.cn (Z.C.); zhangjinbo@njnu.edu.cn (J.Z.)

[2] Key Laboratory of Virtual Geographic Environment, Nanjing Normal University, Ministry of Education, Nanjing 210023, China

[3] Jiangsu Center for Collaborative Innovation in Geographical Information Resource Development and Application, Nanjing 210023, China

[4] Institute of Plant Ecology, Justus-Liebig University Giessen, Heinrich-Buff-Ring 26, 35392 Giessen, Germany; Christoph.Mueller@bot2.bio.uni-giessen.de

[5] School of Biology and Environmental Science and Earth Institute, University College Dublin, Belfield, D04V1W8 Dublin, Ireland

**\*** Correspondence: shenyandai.nnu@gmail.com; Tel.: +86-25-8589-1203; Fax: +86-25-8589-1745

**Abstract:** Soil temperature change caused by global warming could affect microbial-mediated soil nitrogen (N) transformations. Gross N transformation rates can provide process-based information about abiotic–biotic relationships, but most previous studies have focused on net rates. This study aimed to investigate the responses of gross rates of soil N transformation to temperature change in a subtropical acidic coniferous forest soil. A $^{15}$N tracing experiment with a temperature gradient was carried out. The results showed that gross mineralization rate of the labile organic N pool significantly increased with increasing temperature from 5 °C to 45 °C, yet the mineralization rate of the recalcitrant organic N pool showed a smaller response. An exponential response function described well the relationship between the gross rates of total N mineralization and temperature. Compared with N mineralization, the functional relationship between gross $NH_4^+$ immobilization and temperature was not so distinct, resulting in an overall significant increase in net N mineralization at higher temperatures. Heterotrophic nitrification rates increased from 5 °C to 25 °C but declined at higher temperatures. By contrast, the rate of autotrophic nitrification was very low, responding only slightly to the range of temperature change in the most temperature treatments, except for that at 35 °C to 45 °C, when autotrophic nitrification rates were found to be significantly increased. Higher rates of $NO_3^-$ immobilization than gross nitrification rates resulted in negative net nitrification rates that decreased with increasing temperature. Our results suggested that, with higher temperature, the availability of soil N produced from N mineralization would significantly increase, potentially promoting plant growth and stimulating microbial activity, and that the increased $NO_3^-$ retention capacity may reduce the risk of leaching and denitrification losses in this studied subtropical acidic forest.

**Keywords:** temperature change; gross N transformation rates; subtropical acidic forest soil; China; $^{15}$N tracing experiment

## 1. Introduction

Available soil nitrogen (N) affects the growth of both plants and microorganisms. In natural forest ecosystems, available N is supplied via litter and organic matter degradation, plant $N_2$ fixation, and atmospheric N deposition processes; the interplay among these soil N transformations governs the availability of N which is also affected by soil physicochemical and microbial properties as well as local environmental factors (e.g., temperature, moisture) [1,2], where temperature is considered to be a key factor determining microbial activity levels in soils [3–5].

To better understand soil N transformation rates, both net and gross soil N transformation rates should be determined. Although net N mineralization and nitrification rates do provide an indication of N availability in ecosystems, they are not geared towards understanding the dynamics of specific soil N processes [6,7]. When testing the effects of temperature, net rate studies were mainly carried out, often reporting highly variable responses of soil N to temperature changes. This most likely arose because of different temperature responses among transformation processes that are grouped together when considered in the examination of net rates. For instance, net mineralization rates are the outcome of combined effects of mineralization rates—these often derived from different soil organic matter (SOM) fractions which themselves are characterized by different temperature responses—and also the individual consumption processes such as autotrophic nitrification, immobilization, and so forth. Thus, net N mineralization may increase, decrease, or even be non-responsive to changing temperatures [5,8,9]. Only gross rates of soil N transformations can provide the crucial information for insight into dynamics of the internal N cycle between the organic and mineral N pools [10,11].

Some previous studies found differing responses between the net and gross rates of soil N transformation to temperature changes. For example, recent work by Cheng et al. (2015) had suggested that gross rates of N mineralization and immobilization increased in equal proportions with temperature rising from 5 °C to 25 °C in their studied forest soil, leaving the net rate of N mineralization unaltered by temperature change [9]. Earlier, Zaman and Chang (2004) had reported that gross rates of nitrification were balanced by $NO_3^-$ immobilization rates operating at 5 °C to 40 °C, which led to a negligible response of net rate of nitrification to temperature change [12]. Hence, to gain robust process-based insights into soil N availability, how the temperature change affects the individual gross transformation rate must be explicitly taken into account [13,14].

In this study, we investigated the temperature response of acidic forest soils in subtropical China using a $^{15}N$ tracing approach. Current $^{15}N$ tracing methods can simultaneously quantify the gross rates of multiple processes separately, thereby conveying the dynamics of soil N transformation processes more comprehensively and realistically [15,16]. Our aim was to explore temperature response functions for individual gross N rates and to provide an understanding of possible ecological implications in face of ongoing global change.

## 2. Materials and Methods

### 2.1. Soil Sampling

The sampling site was located at the Shuangzhen Forestry Center, Jiangxi Province, China (27°59′ N, 117°25′ E), which is characterized by a typical subtropical monsoon climate. The mean annual temperature is 17.6 °C, with a minimum and maximum monthly average temperatures respectively of 5.6 °C in January and 29.3 °C in July (30-year averages). The mean annual precipitation is 1778 mm (30-year average), of which approximately half occurs from April to June. Dominant vegetation at the site are *Pinus massoniana* Lamb. and *Cunninghamia lanceolata* R.Br. trees The soils here originated from granite and are classified as Hapludults according to the USDA soil taxonomy [17].

Soil sampling was carried out in November 2018. The soil's pH was 5.07, and its organic carbon (C), total N, and water-soluble organic C contents were 19.78 g kg$^{-1}$ soil, 1.92 g kg$^{-1}$ soil, and 236.5 mg kg$^{-1}$ soil, respectively. At the site, five plots (1 m × 1 m) were randomly positioned, in which surface soil (0–20 cm depth layer) was collected after carefully removing the O horizon. All samples were pooled

together, passed through a 2-mm sieve, and then divided into two sub-samples. One was stored at 4 °C for the incubation experiments within two weeks and the other was air-dried to determine soil physical and chemical properties.

## 2.2. $^{15}$N tracing Experiment

A temperature gradient was set up that consisted of five temperature treatments: 5 °C, 15 °C, 25 °C, 35 °C, and 45 °C. There were two $^{15}$N labeled treatments ($^{15}$N-labelled $NH_4^+$ and $^{15}$N-labelled $NO_3^-$) each with three replicates. A total of 120 flasks (250 mL), each containing 20 g (oven-dry basis) fresh soil, were prepared. These flasks were divided into five subgroups (each with 24 flasks) and pre-incubated at 5 °C, 15 °C, 25 °C, 35 °C, and 45 °C, respectively, for 1 day. Then, either the $^{15}$NH$_4$NO$_3$ ($^{15}$N at 10.20 atom%) or NH$_4$$^{15}$NO$_3$ ($^{15}$N at 10.25 atom%) (2 mL) was added evenly to the surface of the fresh soil in each flask, to give respective final concentrations of 30 mg $NH_4^+$-N kg$^{-1}$ soil and 30 mg $NO_3^-$-N kg$^{-1}$ soil. All soil samples were adjusted to 60% water hold capacity (WHC), and all flasks were sealed with a perforated preservative film, after which they were placed into five different incubators for the 6-day incubation. Soil water content in the flasks was maintained by adding water every 2 days to compensate for lost water, especially under the 35 °C and 45 °C treatments. Soil inorganic N was extracted with 2 M KCl (potassium chloride) (soil: solution, 1:5) at 0.5 h, 48 h, 96 h, and 144 h after adding the 15N labeled solution, to determine the concentrations and $^{15}$N abundances of $NH_4^+$ and $NO_3^-$.

## 2.3. Analysis of Soil Properties

Soil pH was determined using a DMP-2 mV/pH detector (soil:water, 1:1.25) (Quark Ltd., Nanjing, China) [18]. Soil organic C was measured using wet digestion with $H_2SO_4$-$K_2Cr_2O_7$ and soil total N by semi-micro Kjeldahl digestion with Se, CuSO$_4$ and $K_2SO_4$ as catalysts. Soil water-soluble organic C was measured with a TOC instrument (Multi N/C, Jena, Germany). Concentrations of $NH_4^+$ and $NO_3^-$ were measured by colorimetry in a continuous flow analyzer (Skalar San$^{++}$, Breda, Netherlands) and their respective $^{15}$N abundances determined by isotope ratio mass spectrometry (Europa Scientific Integra, Crewe, UK) via a modified diffusion method following Brooks et al. (1989) [19].

## 2.4. Calculations and Statistical Analyses

The $^{15}$N tracing model Ntrace was used to quantify the simultaneously occurring multiple gross soil N transformation rates [16]. The data inputted to the model were the concentrations and $^{15}$N excess values—measured $^{15}$N abundance values minus $^{15}$N natural abundance values of $NH_4^+$ and $NO_3^-$. All values entered were means ± standard deviations. Net rates of N mineralization were calculated as the change in inorganic N ($NH_4^+$ + $NO_3^-$) concentration from 0.5 h to 144 h divided by the incubation time, and likewise for net rates of nitrification but using the change in $NO_3^-$ concentration.

Significant differences in the means among the five temperature treatments were tested by one-way ANOVA, carried out separately for each N transformation rate. Based on the actual experimental repetitions, the least significant differences at the 5% significance level (LSD0.05) were calculated for each N transformation rate, which presents the most conservative way to calculate LSDs [20]. The observed error in the observations is linked to the number of actual repetitions and is reflected in the probability density function (PDF) of each parameter [16]. Statistical analyses were performed in SigmaStat 4.0 Analysis.

## 3. Results

### 3.1. Changes in Concentrations and $^{15}$N Enrichments of Mineral N

The modeled and observed concentrations and $^{15}$N enrichments matched well, with $R^2$ values ranging from 0.91 to 0.98 for all treatments (Figure 1). The NH4+ concentration increased with longer incubation times and the higher treatment temperatures. It was significantly highest at 45 °C, indicating

the large effect of temperature on $NH_4^+$ production (Figure 1a). The $NO_3^-$ concentrations remained mostly unchanged, except at 45 °C, where they declined (Figure 1b).

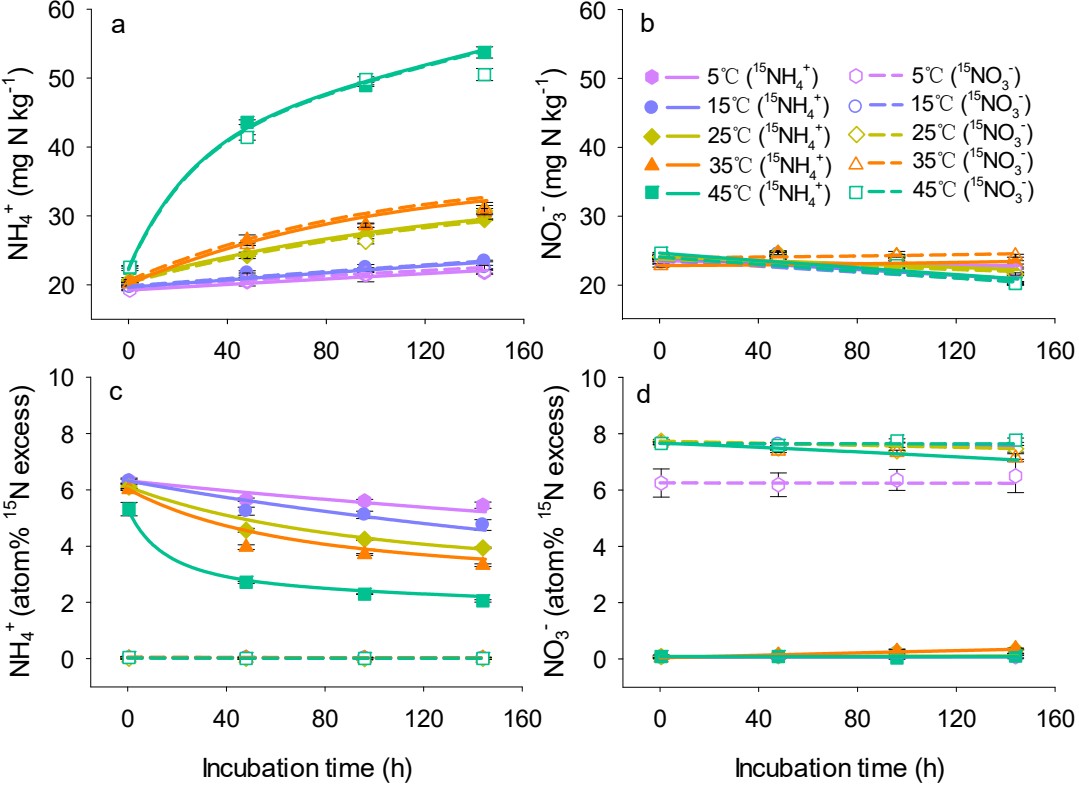

**Figure 1.** Measured (*symbols*) and modeled (*lines*) concentrations of $NH_4^+$ (**a**) and $NO_3^-$ (**b**), and $^{15}N$ enrichments of $NH_4^+$ (**c**) and $NO_3^-$ (**d**). Vertical bars indicate standard deviations of the mean value.

Under the $^{15}NH_4NO_3$ treatment, the $^{15}N$ enrichment in the $NH_4^+$ pool decreased with rising temperature, whereas that of the $NO_3^-$ pool was nearly unchanged (Figure 1c). In the $NH_4^{15}NO_3$ treatment, however, the $^{15}N$ enrichment in both $NH_4^+$ and $NO_3^-$ pools was only slightly changed (Figure 1d).

### 3.2. Influences of Temperature on Net N Transformation Rates

The net rate of N mineralization increased as the temperature rose from 5 °C to 45 °C. The highest rate, 4.3 mg N $kg^{-1}$ $day^{-1}$ soil, was observed at 45 °C, or ca. 25 times that of the 5 °C treatment. Net rates of nitrification in all treatments were negative, and they tended to decline with higher temperatures (Figure 2).

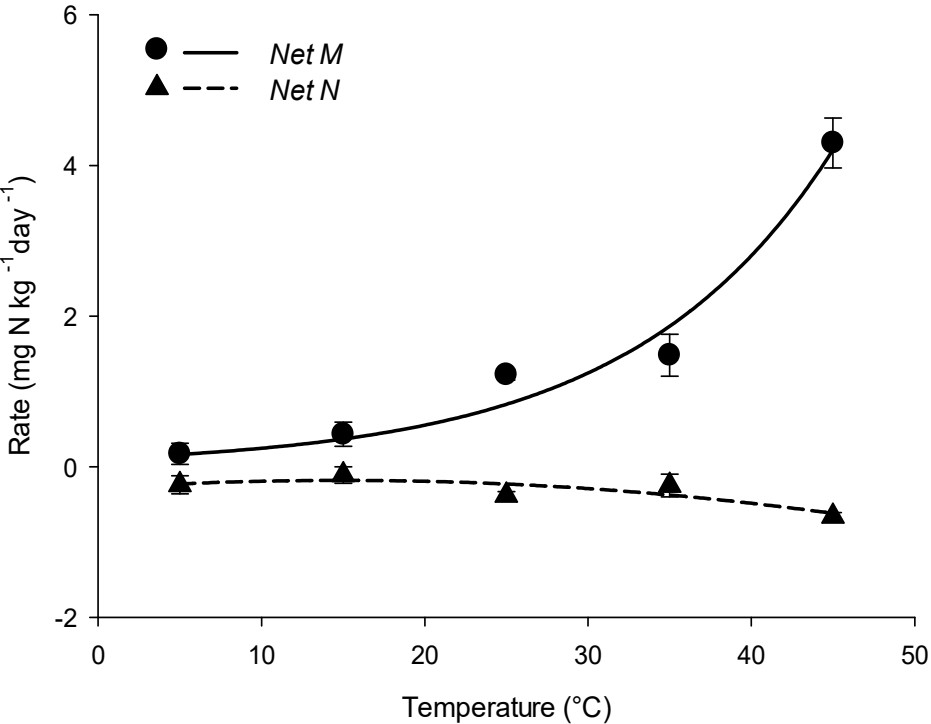

**Figure 2.** The non-linear trends in net rates of N mineralization (*Net M*) and nitrification (*Net N*) as function of imposed incubation temperatures. Vertical bars indicate standard deviations of the mean value.

### 3.3. Influences of Temperature on Gross N Transformation Rates

The gross rates of most soil N transformation processes were significantly influenced by temperature change, except for the dissimilatory $NO_3^-$ reduction to $NH_4^+$, adsorption of $NH_4^+$ on cation exchange sites, and the release of adsorbed $NH_4^+$ (data was not shown, because they were near to zero). The responses of gross mineralization rates of soil recalcitrant organic N and labile organic N to temperature change were significantly different (Figure 3a). The mineralization rate of soil labile organic N pool was evidently promoted by a higher temperature, whereas that of the soil recalcitrant organic N pool was nearly stable from 5 °C to 35 °C but it significantly increased at 45 °C. Consequently, gross rate of total N mineralization exponentially increased with increasing temperature from 5 °C to 45 °C, reaching 5.27 mg $kg^{-1}$ $day^{-1}$ soil (i.e., at 45 °C). Gross rates of $NH_4^+$ immobilization—$I_{NH4}$, i.e., immobilization rate of $NH_4^+$ to recalcitrant organic N + immobilization rate of $NH_4^+$ to labile organic N—were much lower than those of total N mineralization, leading to a net production of $NH_4^+$. The response of gross $NH_4^+$ immobilization rates to temperature change was not as distinctive as the response of total gross N mineralization.

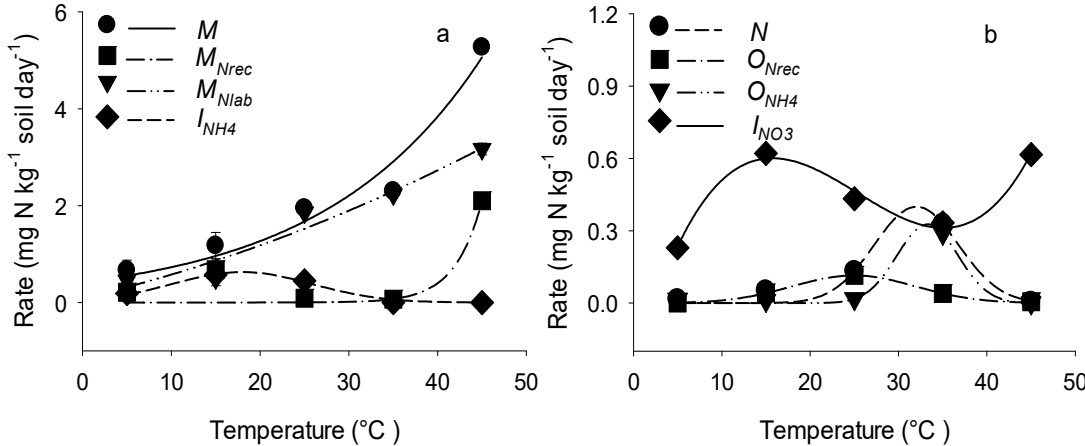

**Figure 3.** Non-linear trends in the gross rates of soil N transformation processes as a function of imposed incubation temperatures. Vertical bars indicate standard deviations of the mean value. (**a**) $M$ ($M_{Nrec}$ + $M_{Nlab}$), gross rates of total N mineralization; $M_{Nrec}$, mineralization rate of recalcitrant organic N pool to $NH_4^+$; $M_{Nlab}$, mineralization of labile organic N pool to $NH_4^+$; $I_{NH4}$, gross rates of $NH_4^+$ immobilization; (**b**) $N$ ($O_{Nrec}$ + $O_{NH4}$), gross rates of nitrification; $O_{Nrec}$, oxidation rate of recalcitrant organic N to $NO_3^-$ (heterotrophic nitrification rate); $O_{NH4}$, oxidation rate of $NH_4^+$ to $NO_3^-$ (autotrophic nitrification rate); $I_{NO3}$, rate of $NO_3^-$ immobilization to recalcitrant organic N.

The heterotrophic nitrification rate was close to zero at 5 °C, but increased from 5 °C through the 25 °C treatment, where it was maximal (0.12 mg N kg$^{-1}$ day$^{-1}$ soil), but then decreased with increasing temperatures (Figure 3b). Autotrophic nitrification was negligible at 5 °C, 25 °C, and 45 °C, peaking at a rate of 0.29 mg N kg$^{-1}$ day$^{-1}$ soil at 35 °C. Therefore, the gross rate of nitrification continuously increased to 0.33 mg N kg$^{-1}$ day$^{-1}$ soil as the temperature rose from 5 °C to 35 °C, but it rapidly decreased to 0.01 mg N kg$^{-1}$ day$^{-1}$ soil at 45 °C. The $NO_3^-$ immobilization rate was significantly higher than the gross rate of nitrification, leading to net $NO_3^-$ consumption. The $NO_3^-$ immobilization rates increased from 5 °C to 15 °C, and decreased thereafter from 15 °C to 35 °C, yet rapidly increased again to 0.616 mg N kg$^{-1}$ day$^{-1}$ soil at 45 °C. Response functions were fitted to describe the how the above dynamics for gross rates of soil N transformation processes were altered along the experimental incubation temperature gradient (Table 1).

**Table 1.** Equations of fitted curves of gross rates of soil N transformation processes as a function of incubation temperature shown in Figure 3.

| Parameter [1] | Equations |
|---|---|
| $M$ | $y = 0.4196 \exp(0.0553x), R^2 = 0.9574, P < 0.05$ |
| $M_{Nlab}$ | $y = 0.0917 + 0.0432x + 0.0006x^2, R^2 = 0.9509, P < 0.05$ |
| $M_{Nrec}$ | $y = 7.2897 \times 10^{-7} \exp(0.3305x), R^2 = 0.8262, P < 0.05$ |
| $I_{NH4}$ | $y = 0.6310 \exp\left[-0.5\left(\frac{x-17.9259}{7.9126}\right)^2\right], R^2 = 0.9800, P < 0.05$ |
| $N$ | $y = 0.4001 \exp\left[-0.5\left(\frac{x-32.0251}{4.7458}\right)^2\right], R^2 = 0.9578, P < 0.05$ |
| $O_{Nrec}$ | $y = 0.1150 \exp\left[-0.5\left(\frac{x-24.9095}{6.9720}\right)^2\right], R^2 = 0.9983, P < 0.05$ |
| $O_{NH4}$ | $y = 0.0093 + 0.7553 \exp\left[-0.5\left(\frac{x-31.7954}{2.2733}\right)^2\right], R^2 = 0.9991, P < 0.05$ |
| $I_{NO3}$ | $y = -0.2863 + 0.1327x - 0.0073x^2 + 8.0250 \times 10^{-5}x^3, R^2 = 0.9834, P < 0.05$ |
| *Net M* | $y = 0.1084 \exp(0.0813x), R^2 = 0.9709, P < 0.05$ |
| *Net N* | $y = -0.2894 + 0.0145x - 0.0005x^2, R^2 = 0.7406, P < 0.05$ |

[1] $M$ ($M_{Nrec}$ + $M_{Nlab}$), gross rates of total N mineralization; $M_{Nrec}$, mineralization rate of recalcitrant organic N pool to $NH_4^+$; $M_{Nlab}$, mineralization of labile organic N pool to $NH_4^+$; $I_{NH4}$, gross rates of $NH_4^+$ immobilization; $N$ ($O_{Nrec}$ + $O_{NH4}$), gross rates of nitrification; $O_{Nrec}$, oxidation rate of recalcitrant organic N to $NO_3^-$ (heterotrophic nitrification rate); $O_{NH4}$, oxidation rate of $NH_4^+$ to $NO_3^-$ (autotrophic nitrification rate); $I_{NO3}$, rate of $NO_3^-$ immobilization to recalcitrant organic N.

## 4. Discussion

Because temperature is a key factor affecting soil microbial activities, it influences microbial-mediated soil N transformations [21–24]. Our results showed that gross rates of individual soil N transformation processes to temperature change responded differently in subtropical acid forest soil; for instance, the mineralization rate of labile organic N pool that significantly increased with temperature rising from 5 °C to 45 °C while the rate of recalcitrant SOM more or less remained stable. Soil enzyme activities are considered limited only by temperature when the supply rate of a substrate exceeds its reaction rate [25]. Therefore, the observation that soil enzyme activities increased with increasing temperature could explain the enhanced mineralization rate of labile organic N pool with temperature. Conversely, the mineralization rate of the recalcitrant organic N pool only increased significantly in the 45 °C treatment, a result possibly explained by the low temperature sensitivity of mineralization of the recalcitrant organic N pool [9]. Previous work done at our study site suggested a low amount of nutrients and a high proportion of recalcitrant compounds characterized the litter of coniferous trees [26]. The temperature sensitivity of microbial-mediated mineralization processes depends on the temperature sensitivity of soil organic matter decomposition [4,6]. Low litter quality likely resulted in low temperature sensitivity for decomposition, thereby requiring a higher temperature for decomposition to finally occur, as it did at 45 °C. The different responses of the gross mineralization rates of soil recalcitrant organic N and labile organic N to temperature change drove the gross N mineralization rates to increase exponentially with rising temperature, a trend that is consistent with many other studies [9,27–29].

Our study found that the gross rate of $NH_4^+$ immobilization gradually increased from 5 °C to 15 °C, a result in line with work by Binkley et al. (1994) and Cheng et al. (2015), yet it declined in the interval of 15 °C to 45 °C, which is likely regulated by the soil C content as observed in other studies [9,30–32]. Higher temperatures could cause labile C to be rapidly assimilated, which resulted in decreasing in C supply to heterotrophic $NH_4^+$ immobilizing microorganisms, thereby further reducing the gross rates of soil $NH_4^+$ immobilization [33–35]. Compared with the gross rate of soil total N mineralization, that of soil $NH_4^+$ immobilization had a less pronounced response to temperature change. With increasing temperature, an exponentially increasing gross rate of soil total N mineralization coupled to the slight response of soil $NH_4^+$ immobilization generated a significant increase net rate of soil N mineralization. Therefore, soil N availability is expected to increase markedly under future global warming conditions, promoting the growth of plants and microorganisms in this subtropical acidic forest soil. Autotrophic nitrification and heterotrophic nitrification are pathways of $NO_3^-$ production in soils [36,37]. Our results revealed functionally different responses of these two processes to temperature change. While autotrophic nitrification is mainly driven by ammonia-oxidizing bacteria (AOB) and ammonia-oxidizing archaea (AOA), heterotrophic nitrification is mainly carried out by heterotrophic fungi or bacteria [37–43]. Previous investigations have suggested heterotrophic nitrification is an important and even dominant pathway for $NO_3^-$ production in acidic forest soil [43–45]. Our results confirm this view, in showing that heterotrophic nitrification exceeded autotrophic nitrification at all temperatures except 35 °C. The former's rate first increased, peaking at 0.12 mg N kg$^{-1}$ day$^{-1}$ soil at 25 °C, then declined with higher temperatures, indicating that heterotrophic nitrifying bacteria or fungi in acidic forest soil preferred a more moderate temperature (e.g., 25 °C in this study), such that their activities may decrease at lower or higher temperatures. Liu et al. (2015) reported an optimum temperature for heterotrophic nitrification of ca. 15 °C in an acidic cropping soil, which contrasted with forest soil (coastal western hemlock) studied by Grenon et al. (2004), for which the heterotrophic nitrification rate was augmented by higher temperature [4,43]. Hence, an optimum temperature for heterotrophic nitrification may be ecosystem dependent and further depend on its organic substrates and microorganisms [46,47]. Our acidic coniferous forest soil featured a high content of complex organic matter, which can stimulate the growth of fungi [9]. Furthermore, Zhang et al. (2019) reported recently that refractory organic C content and fungal gene copy numbers of soil were each positively correlated with heterotrophic nitrification rates [47]. In our study, mineralization

rate of the soil recalcitrant organic N pool significantly increased as the temperature rose from 35 °C to 45 °C, indicating that refractory organic matter content was unlimited, precluding its influence upon heterotrophic nitrification's response to temperature change. The responses of fungi, which are arguably the main divers of heterotrophic nitrification, can determine how heterotrophic nitrification responds to temperature change [39–41,43]. We found a maximal heterotrophic nitrification rate at 25 °C, which is in line with the optimum temperature for fungal growth, 25–30 °C [48,49].

Soil pH is another critical factor affecting the substrate ($NH_3$) and thus capable of greatly influencing the rate of autotrophic nitrification, which research suggests is positively correlated with pH [38,50–52]. In our study, the soil autotrophic nitrification rate was very low and it barely responded to the applied temperature gradient, except for the 35 °C treatment, in which it significantly increased. In acidic forest soils, autotrophic nitrification is carried out by AOA [52,53]. Thus, we suggest the response of autotrophic nitrification to temperature change may have been linked to the temperature sensitivity of AOA in the acid coniferous tree soil we studied. The optimum temperature for AOA is 35–40 °C, so greater autotrophic nitrification at 35 °C is expected, as found in our study [54–57]. Disparate responses of autotrophic vis-à-vis heterotrophic nitrification rates to temperature change led to gross rates of nitrification increasing from 5 °C to its peak at 35 °C (0.329mg N $kg^{-1}$ $day^{-1}$ soil).

The $NO_3^-$ immobilization rate was responsible for net $NO_3^-$ consumption and its response to temperature change substantially affected the available N. It is widely thought that $NH_4^+$ is more easily immobilized by microorganisms than is $NO_3^-$, even if the size of the $NH_4^+$ pool is very small [58–60]. However, our results showed that $NO_3^-$ immobilization rates actually exceeded gross rates of nitrification, especially at 15 °C and 45 °C. This may be a unique feature of these acidic forest soils given similar observations for this kind of system [9,45,60–62]. We showed that $NO_3^-$ immobilization rates accelerated, to a maximum of 0.621 mg N $kg^{-1}$ $day^{-1}$ soil at 15 °C, yet decreased at 15 °C to 35 °C but sharply rebounded from 0.233 to 0.616mg N $kg^{-1}$ $day^{-1}$ under 35 °C to 45 °C. This dynamic response of $NO_3^-$ immobilization rates at different temperatures may be linked to specific responses of different microorganisms (e.g., fungi or bacteria) to temperature change [63].

Since the $NO_3^-$ immobilization rate outpaced the gross rate of nitrification, the net $NO_3^-$ immobilization rates decreased with increasing temperature. This result points to the $NO_3^-$ retention capacity in the studied soil being enhanced by warming, which should reduce the risks of leaching and denitrification losses in this acidic subtropical forest under global warming conditions. However, Jansen-Willems et al. (2016) reported that increased soil temperatures could augment both total inorganic N and $NO_3^-$ pools, mainly due to the gross rates of released stored $NO_3^-$ and total nitrification (autotrophic and heterotrophic nitrification) being promoted by warming [64]. Yet more inorganic N did not markedly enhance $N_2O$ emissions under higher temperatures, mainly because of a reduction in the rates of denitrification and the oxidation of organic N [64]. In our study, however, the responses of $N_2O$ emissions to temperature were not determined. We recommend that future research investigate the effects of changing temperature on $N_2O$ emissions and production pathways and the related microbial mechanisms in acidic subtropical forest soils.

## 5. Conclusions

Our results for a subtropical acidic coniferous forest soil showed that the responses of separate gross N transformation rates to temperature can vary. Because the response of gross rates of $NH_4^+$ immobilization to temperature change was weaker than that of total N mineralization, the net rate of N mineralization significantly increased with rising temperature. The higher $NO_3^-$ immobilization rates than gross rates of nitrification led to higher net $NO_3^-$ consumption that itself decreased with increasing temperatures. These results suggest that, under global warming conditions, soil N availability would significantly increase, which ought to promote the growth of plants and microorganisms, while retention capacity of soil $NO_3^-$ would also increase, which should reduce the risks of leaching and denitrification losses in subtropical acidic forest soil.

**Author Contributions:** Conceptualization, Z.C. (Zucong Cai) and J.Z.; Data curation, X.D., Z.C. (Zhaoxiong Chen), S.D., and X.H.; Formal analysis, X.D.; Funding acquisition, S.D.; Investigation, X.D., Z.C. (Zhaoxiong Chen), and X.H.; Methodology, Z.C. (Zucong Cai), J.Z., and C.M.; Project administration, S.D.; Supervision, Z.C. (Zucong Cai) and J.Z.; Validation, X.H.; Writing—original draft, X.D.; Writing—review and editing, C.M.; All authors discussed the results and commented on the manuscript.

**Funding:** This research was funded by the National Natural Science Foundation of China, grant number 41830642; the CAS Interdisciplinary Innovation Team, grant number JCTD-2018-06; and the "Double World-classes" Development of Geography.

**Acknowledgments:** The study was carried out as part of the IAEA-funded Coordinated Research Project "Minimizing farming impacts on climate change by enhancing carbon and nitrogen capture and storage in Agro-Ecosystems (D1.50.16)" and it was carried out in close collaboration with the German Science Foundation research unit DASIM (FOR 2337). We would like to thank the native English speaking scientists of Elixigen Company (Huntington Beach, California) for editing our manuscript.

**Conflicts of Interest:** The authors declare no conflict of interest.

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
