# Peer review of "Effects of Changing Temperature on Gross N Transformation Rates in Acidic Subtropical Forest Soils"

_forests, doi:10.3390/f10100894_

Round 1
Reviewer 1 Report
Your manuscript is very interesting and should be accepted after a little revision.
1. It is necessary to specify how long the soil samples were stored at a temperature of +4oC before the experiment (line 96). Long-term storage can change the ratio of dominant microorganisms in the soil and, therefore, significantly affect the results.
2. The thesis that heterotrophic nitrification is an important and even dominant route of nitrate formation in acidic soils (line 266 in manuscript) is widespread but not completely unambiguous. For instance, Islam et al. (2007) found that the activity of autotrophic nitrification in acidic soils of agroecosystems can be 2 times higher than the intensity of heterotrophic. This activity may be related to the activity of a group of lithotrophic archaea capable of oxidizing ammonium to nitrite (Konneke et al., 2005)and adapted to low soil pH.
I think it is necessary to take these data into account in the discussion.
3. Figures 1 through 3 if possible needs to be done in the same style (either entirely coloured or black and white).
4. Latin names of tree species should be italicized (line 88)
5. You need to correct typos, for example: line 40 china
line 86 ... and (29.3oC in July...
line 358 ...ttemperature
I am sure that these minimal corrections will not cause you difficulties.
Author Response
Dear reviewer,
Firstly, thank you very much for your time and considerations.
We have revised the article according to your suggestions line by line, except the following comment 3.
3. Figures 1 through 3 if possible needs to be done in the same style (either entirely coloured or black and white).
Author’s response: There are ten treatments in figure 1. We think the coloured lines are looked more clear than black and white in figure 1. So, the coloured style was used in figure 1, while the black and white was used in the other figures.
Reviewer 2 Report
Xiaoqian and the colleagues explored responses of gross N transformation rates to increasing temperature in an acidic subtropical forest soil. Their topic fits well with the scope of Forests journal and will be interested by the readers. This manuscript is well written with clear structure, concise results and concrete discussion. Using the 15N tracing approach, their results showed the distinct responses of recalcitrant and labile organic N to increasing temperature. They also found the changes in the gross rate of heterotrophic and autotrophic nitrification were different. Overall, their results clearly demonstrated that with increasing ambient temperature, the exponential increase in net N mineralization and the unchanged and slightly decreased net nitrification were due to stimulated gross N mineralization and the higher rate of nitrate immobilization. I have the following specific comments which may need to be addressed. Finally, I would like to recommend this manuscript is acceptable for this journal.
Line 29-31, revise this sentence for clarity.
Line 31, but declined at higher temperatures.
Line 36-39, our results suggest that…; and that…
Line 39, in this studied subtropical acidic forest.
Line 65, delete this sentence.
Line 70, …have suggested…
Line 73, rephrase this sentence.
Line 77, move this sentence to the beginning of this paragraph.
Line 78, to explore…
Line 82, Soil sampling.
Lien 86, One more bracket symbol.
Line 95, please clarify what you mean “at the plot level”.
Line 98, here and throughout the manuscript, please adjust the superscripts and subscripts.
Line 116, do you mean water-soluble organic carbon?
Line 145, better to remove ‘significantly’ here as there is no statistical result.
Line 171, where are the data on these processes?
Line 207, rewrite this sentence.
Line 220, This table failed to be explained sufficiently in the Results.
Line 230, as an opening sentence, I would suggest the authors to revise it.
Line 255, rewrite this sentence.
Line 268, rewrite this sentence.
Line 295, do you mean “versus”?
Line 296, led to..
Lien 302, please expand this discussion to make it clearer.
Line 316, As a limitation of the study, you are not meant to present it here. But if you did it, you should to give some speculation about the potential linkage between the response of soil nitrous oxide emission and issues studied here to increasing ambient temperature.
Author Response
Dear reviewer,
Firstly, thank you very much for your time and considerations.
We have revised the article according to your suggestions line by line.
Thanks a lot.
Reviewer 3 Report
Thank you for the opportunity to review the manuscript ‘Effects of changing temperature on gross N transformation rates in acidic subtropical forest soils’. In this study, the responses of gross rates of soil N transformation to temperature change in a subtropical acidic coniferous forest soil were investigated using a 15N tracing experiment with a temperature gradient. This is a very important topic. Despite a lot of previous investigations have reported the effects of temperature on soil N transformation rates, most of them were focused on the net rates of N mineralization and nitrification. Although net N mineralization and nitrification rates do provide an indication of N availability in ecosystems, they are not geared towards understanding the dynamics of specific soil N processes. Only gross rates of soil N transformations can provide the crucial information for insight into dynamics of the internal N cycle between the organic and mineral N pools. Generally, the article is written well. I enjoyed reading this manuscript. It is a very interesting work. The results showed that the responses of specific gross N transformation rates to temperature vary individually, which provided some understanding of soil N transformations under global warming conditions. Thus, I recommend that this manuscript is suitable for publication in this journal after minor revisions.
Line 22: Soil temperature change caused not only by global warming, but also by seasonal variation or other possible factors. Line 26: Should add specific temperature gradient. Line 29: smaller response indicate significant or not significant? Line 32: was should added before not. Line 40: What is the definition of NO3– retention capacity? Line 52: I can’t understand the meaning of “of the last”. Line 67: particular should be changed to individual. Line 74-77: I can’t understand why the authors did this study as the two former researchers have done the similar work. Line 124: should add more information regarding model. Line 279-282: I guess that the future study should focus on the related microbial mechanisms. Line 284-292: should emphasize the limitation of laboratory study.
Author Response
Dear reviewer,
Firstly, thank you very much for your time and considerations.
We have revised the article according to your suggestions line by line, except the following two comments.
Thanks a lot.
Line 124: should add more information regarding model.
Author’s response: The 15N tracing model used in this study had been widely used in the previous studies of gross soil N transformations and the reference has been cited in the article.
Line 284-292: should emphasize the limitation of laboratory study.
Author’s response: The limitation of this study was added in the last paragraph in the discussion section.